## Research Article

tea garden workers; wage insecurity; payment governance; food insecurity; psychological distress

**Corresponding author:**
Mahfuj Rabbani;
Email: rabbanimahi@gmail.com

# Counting pennies, carrying stress: The mental health impact of low wages on tea garden workers in Sylhet and Moulvibazar

Mahfuj Rabbani[1] and Tabassum Islam Jahin[2]

[1]Jagannath University, Dhaka, Bangladesh and [2]Department of Civil Engineering, Sylhet Engineering College Affiliated with Shahjalal University of Science and Technology, Sylhet, Bangladesh

## Abstract

This qualitative study explores how wage and payment practices inflict both economic hardship and psychological distress among tea garden workers in Sylhet and Moulvibazar. Using an interpretivist approach, semi-structured interviews were conducted with 12 purposively sampled workers in Sylhet. Transcripts in Bengali and the local Sylheti dialect were evaluated using reflexive thematic analysis. The findings reveal six key themes centered on the psychological impacts of wage insufficiency, unpredictable payment schedules, disproportionate quota penalties, and opaque deductions. Participants identified low base wages as the primary driver of chronic food insecurity and debt cycles. Furthermore, systemic issues like delayed payments, poor-quality rations, and gendered care burdens severely exacerbated their acute stress. Workers expressed their psychological suffering through culturally encoded idioms such as tension, chinta (worry), and khitkhite (irritability). Ultimately, the study highlights that workers' top priorities for mitigating distress include fair and sufficient wages, transparent payment systems, food security, and accessible healthcare. These findings emphasize that improving mental well-being in this vulnerable population requires systemic reforms in payment governance.

## Impact statement

The research paper has added to the body of evidence on social determinants of mental health in a structurally disadvantaged occupational group in Bangladesh. It establishes worker-articulated channels by which payment control, and not nominal wage rates alone, can play a role in causing psychological distress by recording the perceived experiences of wage and payment practices in tea garden workers. The results are based on the accounts of the participants as opposed to clinical measurement and one should be careful not to make generalisations outside this interpretivist qualitative sample. The research determines perceived pathways and worker-articulated priorities, which are not tested interventions. As far as practical relevance is concerned, the results indicate that there are two categories of response. Short-term (0–6 months) measures that do not need significant resource allocation would be set predictable payment dates, issue itemised wage slips with clear deduction descriptions, reorganise the disproportionate quota-based penalties and enhance the quality and edibility of food rations. Such moves respond to the most common issues expressed by participants. In the long term (6–24 months), they should carry out structural reform in terms of meaningful wage growth in relation to cost of living, institutionalised grievance systems, gender-sensitive support systems (including childcare) and aligning it with national food security and social protection programmes to mitigate the conditions that workers reported were the main causes of distress. The results apply to policy makers because it has been shown that it is not adequate to consider only nominal daily wage rates. Participants felt that dimensions of payment governance such as predictability, transparency and proportionality of deductions were as high as, or higher than the wage level itself, as a source of psychological burden.

## Introduction

One of the largest plantation labour industries in Bangladesh is the tea industry, which has more than 100,000 workers who are concentrated in the Sylhet and Moulvibazar districts. This industry functions under a historically determined plantation regime, where the movement of workers is limited, intergenerational poverty and lack of accessibility to government services (Al Mamun and Gauala, 2025). The workers normally live on estate grounds, are paid on a daily basis and in-kind rations by the estate management, and rely on the estate-managed systems to provide them with basic services. The available literature reports multidimensional deprivation in such communities, such as ill nutritional outcomes in children, insufficient maternal healthcare, high

prevalence of musculoskeletal diseases and healthcare access barriers (Biswas et al., 2020; Iqbal et al., 2020; Kairi and Dey, 2022).

It is on this structural basis that workers in this study reported of how their experiences of payment and wage practices were not only economically constrained but also psychologically burdensome. Particularly, the participants explained how structurally unequal payment systems, where permanent employees earn weekly wages and casual employees earn daily pay, produce qualitatively different income insecurity profiles. They wrote of quota systems of deductions, in which small falls behind daily leaf-picking quotas would provoke penalties at proportions, which did not reflect the minimum wage. Disruption of the normal payment cycles was cited as a source of acute household distress, which resulted in informal borrowing and a decrease in food intake. These descriptions refer to an aspect of the economic life of workers that goes deeper than the level of absolute wage, the mode of governance of wages, by calculation, time, communication and deduction.

The term payment governance is used in this research to denote the entirety of administrative practices according to which the elements of workers compensation are computed, distributed and reported. This includes the time and the frequency of payment; the computation, usage and reporting of deductions; the quality and uniformity of in-kind benefits like food rations and how well and how often workers get clear, readable and challengeable information on the basis of their pay. This is analytically different to the absolute level of wages, yet the two notions are connected to each other: governance failures increase the burden of already-low wages by creating uncertainty, transparency and perceived unfairness to the remuneration system.

There is a substantial amount of evidence in the world, which measures high correlations between poverty levels, material deprivation and common mental disorders (CMDs) in low- and middle-income nations (LMICs), especially depression and anxiety (Lund et al., 2010). Longitudinal and experimental studies have determined that income insecurity, debt, food insecurity and the lack of access to health services are real channels through which poverty causes poor mental health (Patel et al., 2018; Ridley et al., 2020). Clinically significant depression has been reported in the tea garden workers in Sylhet (Nowshin et al., 2021) in Bangladesh specifically, and recent research has described the livelihoods of tea garden workers as structurally marginalised and poverty-trapped (Al Mamun and Gauala, 2025). However, although prevalence research confirms the presence of CMDs within this group, mechanistic qualitative research investigating the perception of certain wage and pay governance behaviours as antecedents of psychological distress, as they are perceived by workers themselves in their daily lived experience, is not found in the literature. A worker-centred qualitative description is necessary to comprehend the experience of payment practices as stress pathways, how these pathways are influenced by gender, job position and employment status, and what alterations workers themselves find possible and meaningful.

This article bridges that gap by using an interpretivist qualitative design within two tea garden districts as the focal point, focusing on worker voices and understanding their explanations of their answers through existing frameworks of occupational health psychology.

## Objectives

The study pursued the following objectives:

1. To understand how tea garden workers perceive low wages and payment governance practices, and the perceived connections between these conditions and psychological stress.

2. To examine how gender, job role (plucking/field vs. factory/processing; casual vs. permanent) and care responsibilities shape workers' reported experiences of wage-related stress, with the acknowledgement that differentiation by employment status and role was only partially achievable within the scope of this sample.

3. To document coping and help-seeking behaviours as described by participants, and to synthesise worker-articulated priorities for change derived from participant narratives, recognising these as participant-defined priorities rather than outputs of a participatory co-design process.

## Literature review

### Income, poverty and common mental disorders

The association between CMDs and poverty and low income is well-supported by evidence that depression and anxiety are significantly related in LMICs (Lund et al., 2010). This comorbidity is often described by social causation: long-term material deprivation increases everyday stressors, undermines coping resources and social isolation and, therefore, increases psychological distress (Lund et al., 2011). Two-way directions are also possible: the impairment of mental health can lower the earning power, establishing poverty in the long term. The recent syntheses have supported the argument of causal mechanisms by proving that the income insecurity, debt, food security and lack of healthcare are feasible pathways between poverty and poor mental health (Patel et al., 2018; Ridley et al., 2020). Psychological effects of poverty seem to be especially high when there is an intergenerational, structurally endorsed deprivation of this nature, that is the conditions that define plantation working environments (Lund et al., 2011; Patel et al., 2018).

### Income volatility, payment insecurity and mental health

In addition to the absolute levels of wages, there is evidence that the predictability of incomes alone is a predictor of mental health. Poor self-reported health and greater psychological distress are linked with insecure compensation schemes such as daily pay, inconsistent earnings and performance deductions, without considering other sociodemographic variables (Thomas et al., 2022). The impact of unpredictable wages on the anticipatory stress and perceived control as the two risk factors of anxiety and the symptoms of depression is evident. This relationship is also supported by quasi-experimental evidence: the introduction of a national minimum wage in the United Kingdom was linked with a lower number of depressive symptoms in low-paid workers (Reeves et al., 2017), and unconditional cash transfer in sub-Saharan Africa had a quantifiable positive impact on psychological wellbeing (Haushofer and Shapiro, 2016). These results reveal that the wage mental health correlation is vulnerable to both income level and income stability and predictability, a correlation that is directly applicable to tea garden.

### Debt, financial strain and psychosocial mechanisms

Poor and erratic salaries often trigger debt and monetary meltdowns. A meta-analysis and systematic review have confirmed that there are strong correlations between personal unsecured debt and poor mental and physical health outcomes (Richardson et al., 2013). Debt is also a psychosocial stressor that results in shame, worry and interpersonal conflict besides restricting future-oriented

planning, in addition to acting as an economic constraint. Longitudinal data show that subjective financial hardship is an indicator of poorer mental health outcomes in part due to the mediating effect of decreased hope and increased shame (Frankham et al., 2020). Such psychosocial intermediaries can be enhanced in low-income workplace activities through the hierarchical power relations, workplace monitoring and an aura of injustice all of which are typical of plantation organisation of labour.

### Food insecurity as a pathway

Psychological distress and CMDs are well-known correlates of food insecurity. Systematic review evidence on developing-country settings suggests that food insecurity, through uncertainty, social anxiety and awareness of deprivation, is linked to worsening mental health (Weaver and Hadley, 2009), and meta-analytic data demonstrate such strong correlations between food insecurity, depression and anxiety among a variety of populations (Pourmotabbed et al., 2020). Low wages, seasonal changes in income and debt cycles are probably the cause of food insecurity in LMIC environments, leading to stress ecologies. Bangladesh studies indicate that there are strong correlations between economic distress and resource insecurity and a high level of depressive symptoms (Hossain et al., 2021). Food insecurity can therefore be discussed as a potential intermediate process where being underpaid can result in the mental ill-health in tea garden settings.

### Occupational stress models

There are three theoretical frameworks that shed light on the mental health implications of wage conditions in a work environment:

a) Conservation of Resources (COR) theory: COR theory states that stress is caused by actual or threatened loss of valued resources, whether material (money and food), social and psychological (security and esteem) and that stress is aggravated when people have no means of restoring lost resources (Hobfoll, 1989). Poor wages are a structural shortage of resource and inconsistent remuneration establishes a continuous fear of additional loss. COR is anticipated to forecast high levels of distress especially in situations where structural circumstances constrain the same resources required to cushion stress.

b) Job Demands-Resources (JD-R) model: The JD-R model suggests that exhaustion and disengagement – antecedents of poor mental health – are the result of the failure to balance high job demands (e.g., physical labour, quota pressure psychosocial pressure) with sufficient job resources (e.g., autonomy, social support and fair pay) (Demerouti et al., 20 physically taxing plantation labour and poorly and unreliably paid are an example of a high-demand, low-resource set up.

c) Effort-Reward Imbalance (ERI): ERI theory theorises work-stress as a result of perceived lack of balance between high effort and low reward, operationalised in terms of wages, esteem and employment security, with the resultant strain being long-lasting physiological and psychological. Future meta-analysis data prove that chronic exposure to ERI correlates with the risk of getting depressive disorders (Rugulies et al., 2017). The lack of structural ERI is not accidental to low-wage labour but it is inherent to its organisation.

### Tea garden workers in Bangladesh: Existing evidence and research gaps

Previous studies that have been conducted in Bangladesh have reported prevalence rates of depression in tea garden workers in Sylhet and have identified sociodemographic factors associated with CMDs, which proves that CMDs are a measurable problem of the population health (Nowshin et al., 2021). Similar structural deprivations were also documented in similar studies, such as poor nutrition status of the children of the workers (Iqbal et al., 2020), high rates of musculoskeletal symptoms (Kairi et al., Kairi and Dey, 2022) and service access inequalities across multiple dimensions (Biswas et al., 2020). Precarious livelihoods and structural marginalisation have also been brought into the limelight of qualitative and livelihood-oriented research (Al Mamun and Gauala, 2025; Gupta et al., 2025; Islam and Al-Amin, 2026).

Two fatal gaps are still there, however. First, although prevalence studies confirm that CMDs are present in this group, no mechanistic qualitative research has been done on the perceived role of the particular wage governance practices in this group such as quota-based systems of deductions, payment timing, punitive penalties and transparency failures in causing psychological distress, based on the daily lived experience of workers. Second, the quantitative research that exists fails to capture the subjective aspects of payment practice such as perceived injustice and the culturally expressed forms of distress that employees express. Both gaps have been directly addressed in the present study.

### Conceptual model and theoretical framework

The research is supported by a sensitising conceptual framework used not as a causal model to test, but as a conceptual framework on how data will be collected and how to interpret themes. The model combines three different theoretical approaches: ERI, JD-R and COR.

The contextualisation of work-related stress by ERI is based on the alleged lack of balance between high effort and low reward, which is not limited to wages alone but extends to esteem and employment security as well (Siegrist, 1996). JD-R suggests that strain occurs when the high job demands such as quota pressure, physical workload is not offset by the adequate job resources such as fair remuneration, supervisory support and worker voice (Demerouti et al., 2001). COR conceptualises low wages and intermittent payment as chronic resources inadequacies, which are more destabilising in the situation where households cannot restore basic resources (Hobfoll, 1989).

Such occupation stress processes are contextualised in the poverty-related processes, especially financial strain, debt cycles and food insecurity, which are considered possible mediating pathways between wage and payment conditions and psychological outcomes (Lund et al., 2010; Richardson et al., 2013; Pourmotabbed et al., 2020). Other psychological processes, such as loss of hope, shame and sense of injustice, can increase distress in further proportions in case of chronic deprivation (Frankham et al., 2020; Ridley et al., 2020).

The framework will therefore place both wage level and the quality of payment governance as structural determinants that influence mental health of workers in two mutually reinforcing ways: an economic pathway (financial strain, debt and food insecurity) and an occupational-psychological one (ERI: perceived injustice of disproportionate deductions; JD-R: resource deficits of low autonomy and lack of grievance). It is expected that

moderating factors, including gendered care burdens and employment insecurity will magnify these pathways by exposing people more and limiting available coping resources. Cross-cutting the six analytic themes: Theme A is similar to COR (resource loss) and the economic pathway; Theme B to ERI (payment shock as effort reward disequilibrium) and JD-R (acute demand surge without buffering resource); Theme C to ERI (perceived procedural injustice) and COR (multiplicative resource threats) and Theme D to JD-R (gendered demand overload.

Where the results indicate a cyclical nature, such as where distress continues to limit earning capacity and resource access, this would be explained with the two-way logic of poverty-mental health associations proposed by Lund et al. (2011) and Ridley et al. (2020), but not as a causal loop that has been empirically demonstrated in this study.

## Methodology

### Study design and setting

The research design of the study was qualitative based on an interpretivist paradigm, which is suitable in exploring subjective meanings that workers place on wage and payment experiences and how they think these factors impact their daily wellbeing (Creswell and Poth, 2018). The study was carried out in 2024–2025 (the exact year to maintain the confidentiality of the participants) in tea garden villages in two districts, Sylhet and Moulvibazar, which comprise the majority of the tea garden labour force in Bangladesh. The work in tea gardens is often identified by position (plucking and field work vs. factory and processing) and by type of employment (permanent vs. casual/non-permanent), arrangements that directly influence the frequency of wages, quota demands, entitlement to deductions and access to benefits, which are the focus of the objectives of the study.

### Participants, sampling strategy and sample size

The sampling method was purposive and maximum-variation, and it was intended to be heterogeneous covering three dimensions; (i) gender, (ii) job role (plucker/field worker vs. factory/processing staff) and (iii) employment type (permanent vs. casual/non-permanent) (Patton, 2015). These dimensions were chosen since the research questions focus on the difference in wage and payment experiences based on gender, role and responsibility of care.

The main sample of analysis was formed by 12 participants in two interview forms in Sylhet. The former format was one-on-one and small-group interviews (n = 6 respondents in various sessions): one individual interview (Toma Goyala), one group interview where Smriti Goyala along with two other respondents (Mohon and Raju) participated together (the interview) and two additional one-on-one interviews. Small-group interview or triadic interview is a semi-structured interview conducted with two or three participants, which gives peer validation and natural conversational dynamics in a structured interview. The second form was a group discussion of six people in Sylhet. The analytic sample is made up of all 12 of the participants.

Another group interview was then carried out in Moulvibazar to triangulate contextually only. The purpose of this interview was to contrast wage and payment rates between permanent and non-permanent employees in a second site, which would enhance the contextual meanings of the key results. It is not included in the analytic sample (n = 12); the number of participants in the Moulvibazar session were not counted in the field notes when they were systematically and were not analysed using the same systematic methods used on the primary sample. This exclusion can be justified by the fact that the interview of Moulvibazar was contextual and confirmatory rather than primary analytic and its inclusion in the sample would have confused the methodological basis of the primary analysis.

Recruitment was done until an operationally defined data saturation was achieved a concept where new conceptual information and emergent themes were not obtained during consecutive interviews (Guest et al., 2006). Practically, there was a consensus that the saturation point was achieved through the last group interview in Sylhet, where no new themes that were substantively new were reported. The information power principle also informed stopping decisions, where the research question is narrowly defined, the sample is occupationally homogenous and the data are rich in dialogues (Malterud et al., 2016). The small sample restricts the extent of representation that can be made and should be recognised.

About 20–25 employees refused to take part in the recruitment. This refusal rate is publicly recognised, those who agreed to participate may be a smaller group of workers more willing or able to express grievances and may therefore have introduced selection bias (an aspect of over-representing certain experiences such as acute distress or articulate dissatisfaction, and underrepresenting others such as those who fear retaliation, less engaged or more alienated to the research processes) (Tong et al., 2007). This is a material constraint of the scope of the study.

### Data collection procedures

The interviews were all in the Bangla and the local Sylheti language, which was chosen according to the language preferences of the participants and also so as to allow them to express the distress in a culturally appropriate manner. The actual time of individual in-depth interviews was 30–60; the group interview was about 60–75. The semi-structured interview guides covered: (1) wage and payment practices, including quota targets, types of deductions and payment delays; (2) perceived effects on stress and mind–body wellbeing, using vernacular language, for example, chinta (worry), mon kharap (low mood) and khitkhite (irritability); (3) gender, job role and care responsibility variation and (4) interviews were done privately where feasible. The use of clinical terminology and diagnostic framing was avoided.

To obtain data, audio recording (with a written informed consent) was used to capture data, field notes were taken at the time of data collection (of non-verbal cues and situational observations), and reflective memos were created at the end of the interview (of emerging themes, and signs of distress in the participant).

Transcription and translation: Transcription and translation were done verbatim in original Bangla or Sylheti dialect. The research team members who are not only fluent in both the English and the Bengali languages translated the transcripts into English. In order to maintain the emotive tone and idiomatic texture of the words of the participants, especially culturally embedded idioms like tension, chinta and khitkhite, a second team member cross-tabulated translated excerpts with the original transcripts. Direct English translation poses the threat of defamiliarising the original wording, so vernacular words are left in quotations with English glosses in parentheses.

## Ethical considerations

Sensitivity of such issues as financial hardship, mental strain and family conflict was discussed by ethical procedures. Participation was voluntary and informed consent in writing was taken before data collection. Participants were made aware of their right to not answer any question and at any point leave the study without any repercussions and they signed a separate consent to the audio recording and to the use of anonymised quotations. Because of the possible distress caused by talking about economic suffering and emotional burden, the interviews were carried out in non-stigmatising terms with the possibility to pause or withdraw, and the information regarding the possible community support and provision was offered. The procedures of confidentiality involved deletion of the identifying information in the transcripts and notes, provision of pseudonyms to the participants and limiting access to the research team. Confidentiality in group contexts was not assured and the participants were asked to keep the group discussions confidential.

## Data analysis approach

Reflexive thematic analysis was used to analyse the data (Braun and Clarke, 2006) and the process was completed in six stages: (1) familiarisation through repeated reading and listening, (2) preliminary code generation, (3) searching themes, (4) reviewing themes, (5) defining and naming themes and (6) writing a report. The study objectives and theoretical framework informed deductive sensitising codes (e.g., payment irregularity, quota pressure and gendered care burden) and inductive codes were based on the language of the participants (e.g., worry about food, sleep disruption and irritability). Taking a subset of transcripts and independently coding them by a second researcher did not attempt to determine inter-rater reliability in a positivist sense, but served to strengthen reflexivity through the identification of divergent interpretations and the generation of discussion about the meaning and scope of codes (Nowell et al., 2017). The emerging interpretations were tested on raw data and field notes periodically through peer debriefing (Shenton, 2004).

## Trustworthiness

Credibility, transferability, dependability and confirmability were used as the criteria of trustworthiness (Lincoln and Guba, 1985; Shenton, 2004).

- Credibility: cross-method triangulation (individual, triadic and group interviews), use of verbatim quotations in original-language terms, peer debriefing and reflexive memoing.
- Transferability: detailed description of location, work roles and employment types to support analytic inference to comparable tea garden settings.
- Dependability: maintenance of an audit trail covering instrument versions, recruitment documentation (including refusals), coding iterations and theme revisions.
- Confirmability: documentation of reflexive assumptions and grounding of all interpretations in data excerpts and field notes. Reporting transparency was further supported through compliance with COREQ guidelines for interviews and focus groups (Tong et al., 2007).

## Findings

Six themes were identified. Themes A–D deal with Objectives 1 and 2 (the perceived impacts of wage and payment practices, and gender, role and employment status variance). Theme E documents represented and cultural manifestations of distress. Theme F gives worker-articulated change priorities (Objective 3 synthesised based on participant narratives). This is indicated where frequency of expression among participants can be approximated. Field observations are clearly defined as contextual data and not regarded as analytic evidence of mental health outcomes.

### Theme A: Wage insufficiency and daily survival trade-offs

Stated by the majority of the participants during individual and group interviews. Wages were always described by workers as structurally low to cover the basic living expenses. The overwhelming experience was not the budgeting in a wage but the ongoing triage between the need to be postponed: the diversity of food, healthcare, education of children, repair of houses and debt repayment. Toma Goyala wrote of the impossibility of the household sustaining itself on the joint upkeep of wage earnings, and how the weekly payment regime has been brought to stabilise a regime of anxiety about unmet need:

> I do not make enough due to the income of my husband… How will we survive? The children have to be fed every week… With this wage it doesn't work… When they pay instalments, there is nothing to purchase to last the entire week, there are no vegetables, there is no rice. (Toma Goyala, pseudonym)

The concession of food was over and over again not called an extraordinary suffering, but was a commonplace aspect of the money making process. This was simply stated by Smriti Goyala:

> That is impossible… At night, we do eat once only sometimes. (Smriti Goyala, pseudonym)

Subgroup comparison: Wage insufficiency was more severe in cases of casual workers and those that had care responsibilities. Toma described her cessation of formal wage working when she had children and resorted to informal income earning (conducting a ring game) – how low wages and lack of childcare services can lock women out of formal jobs. Casual workers always claimed to be more vulnerable than permanent workers due to the lack of a regular cycle of an income per week, which produced qualitatively distinct uncertainty images. Multiple participants reported that an implicit cycle of a hard week (reduced food consumption [plain rice only]) and informal credit to get them through between paydays was a normalised experience of scarcity, not an aberrant experience.

### Theme B: Payment timing, quota rules and acute stress cycles

Eminent in group and individual interviews especially among plucking and field workers. The respondents explained that the methods of payment calculation and quota structures created foreseeable spikes of acute stress. The weekly cycle of payment was in itself perceived to be chronically anxiety generating since household demands are daily and wage earnings are weekly. Smriti had outlined the quota system, a daily minimum of 23 kg of tea leaves, and the disproportionate system of penalty on failure:

> The price of three fewer taka per kg having been received, then, subtracting 1 kg of this out of our salary, they cut 9 taka per kg… Our

attendance is 23 kg of leaves… They trim 10 kg leaves… in case it is even 1 kg short of it… (Smriti Goyala, pseudonym)

This system of penals, in which lessening by even 1 kg caused a loss at a proportion not in keeping with the base wage, was seen by the players as extremely unfair. Toma also explained that a missed work day would lead to doubled losses due to deduction of wages and withdrawal of ration both at the same time:

Today, either I have to go or they will register the 1200 of the week that they will take out, and they will not pay the ration either (Toma Goyala, pseudonym)

In order to put the wage calculation system into perspective to readers: a daily quota of 23 kg was mentioned by the participants against which a base daily wage was paid. Lack of quota had the effect of encouraging a deduction not at the base per-kg wage rate but at a higher penalty rate (stated to be 9 taka per kg below quota), so that small under-achievements produced a large wage loss proportional to the amount of the shortfall. This hierarchy was felt to be reproving and undisputable. A good week was one where there was no illness or emergency to miss quota; a hard week was one where sickness or failure to work or quota deficit had set the ball rolling through compounded wage and ration losses into emergency borrowing a process that many respondents said was predictable but unavoidable.

### Theme C: Deductions, ration quality and perceived unfairness as psychological pressure

Existing in individual and group interviews. This theme summarises two analytically different but related sources of misery: the build-up of several types of non-performance-related deductions, and the inadequacy of the quality of in-kind ration. They are discussed individually here to explain their different roles in perceiving unfairness among workers.

On the deduction side, Komol (group interview, Sylhet) explained the psychological impact of numerous overlapping deductions used on top of the deductions related to the quotas as explained in Theme B:

Plant a tree, they chop money; want to keep cows, they chop money… now they started to collect electricity bills… all this leads the mind to the mental pressure, as well (Komol, pseudonym).

Throughout the ration side, Toma was able to explain the in-kind rations as not only poor quality but also something that caused physical injury:

They bring nothing but flour… it is not edible… it is red… it creates stomachache… not all can eat it… they have stomach issues. (Toma Goyala, pseudonym)

A gap between the portrayed image of support of workers and the reality of workers increased the perceived unfairness. Parul emphasised the discrepancy between external imagining of the ration system and actual provision that is obtained:

According to the foreigners, the idea of rationality implies rice, lentils, potatoes all that and the fact that we have medical care. Things are not so realistic, though. The picture is similar in every garden.

The obscurity of the inexplainable deductions, and the insufficiency of in-kind assistance, had been felt as increasing the iniquity of meagre pay, with a procedural and distributive injustice to an already restricted material condition.

### Theme D: Gendered burdens, household pressure and emotional strain

It was mainly based on testimonies of the women participants; the comparison of casual and permanent women workers was not systematic due to the samples. Even though the participants noted that nominal wage rates between women and men are equal, gendered burdens were manifested in the unequal distribution of unpaid domestic and childcare tasks and not in the form of official wage disparities. This topic will cover Objective 2, although it may be partially partially realised in terms of role- and employment-based differentiation, as mentioned above. The story of Toma could depict how childcare restrictions alter income schemes, as well as psychological health. She had no choice than to resort to legally risky informal income earning activities denied the choice of returning to formal garden employment because of the lack of affordable childcare:

I began this game with the ring… I had no choice… I am alone in my home… where will I leave my two children when I go out to work?… I cannot see my children starve to death. (Toma Goyala, pseudonym)

Smriti explained the continuous, inexorable quality of her anxiety as spilling out of the workplace into domestic life, where wage anxiety met with social and moral anxieties about the future of her daughters:

At the workplace I am thinking of work, and when I am home I am thinking of home and how to educate my children in such a way that no one will judge them in society… I have three daughters… and all day it is a strain… (Smriti Goyala, pseudonym)

This intergenerational aspect, the interest not only in survival, but in the education of children and their social standing, was a common thread in the stories of women workers and added to the economic pressures of the moment in Themes A and B.

### Theme E: Embodied and cultural expressions of distress

Expressed by most participants across interview formats. Workers consistently used culturally embedded idioms rather than clinical diagnostic terminology to describe their experiences. The primary idioms were tension (tension), chinta (worry), khitkhite (irritability/ being easily annoyed) and somatic complaints including sleep disturbance, headaches and body discomfort. These terms are treated as emic descriptions of psychological and somatic experience and are not interpreted as clinical indicators.

Toma described tension as a visibly embodied state inseparable from material circumstances:

There is tension there… you can see who is tense… it is about food and it is about so many things. (Toma Goyala, pseudonym)

Komol connected economic insecurity directly to embodied care-related worry when a wage delay prevented him from accessing healthcare for a sick child:

My son is ill I cannot get him to the doctor… I am straining on whether I will receive the salary and take my child to the physician… I am in deep worry. (Komol, pseudonym)

Khitkhite – a Bangla term connoting a state of easy irritation or heightened annoyance in response to sustained pressure – was described by Komol in the group interview as a consequence of persistent strain:

Yes, many times I am annoyed/ irritated. (Komol, pseudonym)

Contextual field note [observational data]: The research team observed that approximately 20–25 workers declined recruitment and several appeared visibly frustrated and disengaged during initial approach. This observation is recorded here as contextual information pertaining to the emotional climate at the site during data collection and is not offered as analytic evidence of specific mental health outcomes. It does, however, suggest that the accounts gathered

may reflect a broader pattern of community weariness that could have shaped both the willingness and manner of participation.

### Theme F: Worker-articulated priorities for change

Made based on individual and group interviews. The priorities of workers were tangible, experience-based, and always focused on wage increments, food security and open payment habits. They are offered as participant-articulated priorities based on thematic synthesis rather than the results of a participatory co-design process. Wage increment: Parul also expressed a certain amount of the daily wage that she considered as a survival necessity: The pay is to be 500 taka daily. Smriti conditioned the value of nominal wage growth on a structural observation of cost-of-living processes:

> If daily 500… then we could manage… However, as much as wages keep rising, prices also rise in the market hence it might not help a lot… We could live, had the price of rice, potatoes, salt, been kept down. (Smriti Goyala, pseudonym)

Food security as a main mental issue: Smriti articulated the underlying character of food access as mental health: The desire is expressed as follows;

> I want us to be able to eat well three times a day. (Smriti Goyala, pseudonym)

Open and equitable payment: Several participants expressed how they wanted more detailed descriptions of deductions and more holistic reimbursement programmes. Mohon Pashi (contextual interview, Moulvibazar – to illustrate a point made in the priority accounts of the workers) explained the system of partial medical reimbursement:

> One must pay oneself first… and then he can pass the bill and give them money… my cost was 5,000 and they would give me 2,000. (Mohon Pashi, pseudonym)

Support of external assistance: Some respondents remembered the intermittent assistance of organisations, such as BRAC, a local NGO, called Usha, but described it as informal instead of structural:

> BRAC was established before… two of my children studied BRAC school until class five… Usha taught our children dance and music… during COVID they made donations… and clothes during puja (Komol, pseudonym).

### Discussion

#### Theoretical integration

The descriptions of the participants in all six themes can be interpreted in terms of ERI, JD-R and COR, but the results are presented as the indications of how the workers perceived and experienced those dynamics, rather than as the proof of the causal processes.

ERI throws light on the fact that distress was not just manifested in the form of suffering but in the form of feeling of unjust exchange. Employees felt that physical labour was heavily rewarded with poorly paid wages that were unreliably deducted and often long delayed (Siegrist, 1996). The disproportional quota penalty regime, in which a 1-kg deficit led to a loss more than the base wage per kilogram, is a direct implementation of the ERI construct of the perceived inadequacy of the reward in the form of procedural injustice. Future meta-analytic data validate high risk of depressive disorders with long-term ERI (Rugulies et al., 2017). More importantly, poor reward was seen on many levels: on-time, openness and dignity of the deduction procedure, not just on wage level.

JD-R sheds light on the working set up where every day experiences were subjected to as psychological load. Respondents reported an enduring imbalance; high demands (quota pressure, physical labour, supervisory control and domestic care demands) and continuously low resources (little autonomy, ineffective or non-existent grievance process, poor financial buffers and weak voice). Under this set up, the JD-R model predicts exhaustion and disengagement (Demerouti et al., 2001; Bakker and Demerouti, 2007). The narratives of humiliating supervisory experiences and disregard of complaints by the participants are quite appropriate in the JD-R framework because demeaning supervision and limited voice drain the relational resources otherwise used to mitigate occupational strain.

The most consistent explanation of the cyclical, compounding nature of distress is COR. Patterns of hard week described by workers (where workers do not earn enough money to live, so they borrow it, and this diminishes their social capital and dignity and their ability to cope) are also aligned with the COR concept of resource loss spirals (Hobfoll, 1989). Quota failures and deduction events served as antecedents to various concomitant resource threats (financial, nutritional, social and psychological) and produced distress responses, which workers perceived as cumulative and mounting.

#### Alignment with existing literature

The results of the study are consistent with macroeconomic and occupational data demonstrating the connection between economic insecurity and mental health. The trend of fluctuation of payments and acute stress circles is echoed in the population-wide research that income variability is an independent predictor of depressive symptomatology (Prause et al., 2009). The discourse normalisation of trade-offs and credit buying between workers in the food sector is a parallel to the loss spirals inherent to COR where food insecurity serves as the predictive factor and the reinforcing factor of psychological distress (Pourmotabbed et al., 2020).

The motif of unclear decisions and a sense of injustice is also in line with the organisational justice literature that records correlations between low perceived procedural justice and psychological distress (Kobayashi and Kondo, 2019). Unjustified deductions and unjustified dismissal of complaints are the signs of procedural unfairness, which makes ERI even more complex as the rewards become not only low but also randomly and unfairly distributed. The testimonies of humiliation and rejected grievances of the participants also resonate with the evidence that the exposure to degrading workplace dynamics correlates with worse mental health outcomes (Verkuil et al., 2015); however, the plantation setting is not structurally the same as the formal sector workplaces that are typically researched in this literature.

The results concur with reported multidimensional deprivation in tea garden populations (Biswas et al., 2020; Gupta et al., 2025) and with the epidemiological observation of high levels of depression among Sylhet tea garden workers (Nowshin et al., 2021). The current research offers qualitative and mechanistic description of the perceived conditions, which can result in such prevalence estimates. The intergenerational aspect, which is the perceived interest of the workers in the education and social prospects of children, is consistent with the fact that chronic poverty is recorded by Islam and Al-Amin as a cause of intergenerational educational inequality in tea gardens (2025).

## Comparative perspective: South Asian plantation labour

This psychological pressure recorded here is not peculiar to Bangladesh but seems to be a structural feature of South Asian plantation labour regimes in general. The situation observed by Ashwini et al. (2017) in Tamil Nadu, India, where high workload, low wages and socially confined settings were perceived as the causes of psychological discomfort, recorded a high mental health burden among tea estate workers, which is a combination that is highly similar to that of the current study. Studies on plantation labour in Sri Lanka have also drawn attention to the overlap between poverty, physical morbidity and mental suffering within a community paradigm of residential enclosure, restricted locomotion and hereditary occupational placement, just as is the case in the intergenerational plantation system in Bangladesh. The similarity of stress-inducing factors across national settings, namely the physical task demands, the perceived lack of reward, opaque governance and limited worker voice, is indicative of the fact that these factors could be structurally ubiquitous features of the plantation-style of labour organisation in the region, as opposed to being features of the Bangladesh regulatory environment. This regional homogeneity facilitates the plausibility of the identified pathways between wage and mental health and suggests that a cross-national comparative study of plantation labour mental health would be a desirable direction of future research.

## Variation across participant characteristics

The results indicate that there is significant non-homogeneity in the workforce, and tea garden workers should not be viewed as a homogenous group. Women who had care responsibilities were twice burdened: formal labour combined with unpaid domestic and childcare pressures, limited their ability to work formally and reduced their coping possibilities (Theme D). The stress of casual workers was more strongly conditioned by the unpredictability of income than permanent workers, paid on a weekly basis instead of daily; permanent workers also had the weekly cycle as a stressor because of the discrepancy between daily dependence and weekly income. Plucking and field workers were the most immediately strained by quotas, deducted, yet factory workers had similar restrictions of mouth, access to grievances and perceived management equity. These differences suggest that interventions cannot be applied to all employees similarly but instead should be differentiated based on their employment type and role.

## Policy and management implications

The results suggest that the payment quality, including regularity, transparency, deductions of the same proportion and predictability of such deductions, must be a major focus of the wage policy in the tea industry rather than a secondary issue when nominal wage rates are discussed. According to ERI and COR, the perceived injustice and resource threats are compounded by disproportional penalty arrangements even at the same level of absolute incomes (Hobfoll, 1989; Siegrist, 1996). Nominal base wage rates can be used to mask the sources of distress at governance level.

At the level of the estate management, JD-R shows that, despite the slow structural wage reform, aimed changes in job resources may partially relieve strain (Demerouti et al., 2001). The estate managers are mandated to adopt respectful and transparent communication between the managers and the workers, regular and documented deductions, effective grievance procedures and employee feedback technologies. The labour inspectors and enforcement officials have the responsibility to ensure minimum wage is complied with and that deduction practices are observed to be proportional and transparent. Government agencies and labour policy makers are in a position to enforce itemised wage slips, controlled deduction systems and gender sensitive workplace benefits including childcare subsidies. Intersectoral government collaboration would be concerned with the connections between the estate-level support and national food security and social protection programmes.

The economic insecurity, food compromise and embodied distress found in these narratives indicate that wages cannot be used to address mental wellbeing, integrative reactions between labour conditions and food security and social protection are also in line with the larger evidence base of poverty-mental health (Ridley et al., 2020).

## Recommendations grounded in findings

### Short-term, estate-level actions (0–6 months)

- itemised wage slip and deduction documentation: Based on the description of Komol and Parul in regard to unexplained accumulating deductions (Theme C). The perceived procedural unfairness would be directly tackled by itemised post-deduction wage slips and a documented policy of deductions, which includes a mechanism of same-week correction of any contested deductions.
- Determined and told payment disbursements: Based on the testimonies by Toma and Smriti about the weekly payment cycle as one of the anxiety producers (Theme B). Predatory announcements of payment dates and written announcements of delay would aim at income volatility (Prause et al., 2009) and COR-identified resource threat (Hobfoll, 1989).
- Reform of proportional quota deduction: This is based on the idea of disproportional penalties on small quota underperformances described by Smriti in the description of the quota system (Theme B). Procedural unfairness recorded in Theme C would be solved by redesigning the calculation of the deduction to be proportional to the base wage rate. Dignity-oriented training of supervisory communication: Based on the testimonies of the participants concerning the humiliating experience and denied complaints. Exposure to relational resource depletion would be minimised by short-term training on respectful and non-humiliating performance feedback (Verkuil et al., 2015).

### Structural reforms (6–24 months)

- Institutionalised employee representation and grievance procedures: Rationalised by the reports of participants of the lack or ineffectiveness of voice structures (JD-R resource deficit). This structural gap would be filled by formalised grievance processes that have documented results (Bakker and Demerouti, 2007).
- Gender-sensitive support services: Based on the testimonies of Toma and Smriti about the limitations of childcare and gender-related care-related burdens (Theme D). Childcare on-site or with a subsidy, healthcare access subject to maternity and overt anti-humiliating policies on gender basis are justified.
- Better quality of the ration and nutritional sufficiency: It is based on the description of inedible and poisonous ration by Toma (Theme C). Enhancing the quality of rations would help to depreciate one channel by which food insecurity mediates distress on wages (Pourmotabbed et al., 2020).

• Connection with food security and social protection programmes: Since food insecurity is tightly connected with the psychological distress in the responses of participants, integrating with local food security and social protection programmes would decrease downstream psychological damage as structural wage reform continues.

## Study limitations

There are a number of limitations that should be mentioned openly.

First, the small primary analytic sample (n = 12) severely limits the scope of the diversity of tea garden workers, which can be sampled at Sylhet and Moulvibazar. Although the information power principle provides the analytic depth to this particular research question, in a narrow occupational sample (Malterud et al., 2016), these findings cannot be extrapolated to a bigger picture of the interpretivist qualitative research.

Second, the over-reliance on a few voices, and those voices in particular are Toma Goyala and Smriti Goyala, the most detailed and best documented in the data. Such concentration is dangerous because it may exaggerate some experiences and views and underrepresent others, especially those of the workers whose conditions, temperaments or modes of communication do not match these central actors.

Third, fear of retaliation was likely to limit the disclosures of participants. In plantation environments, where the ability to speak freely is mediated by the employer, even with promises of confidentiality, employees who do not have residential housing attached to their jobs and access to basic services are at very real risk of reprisal due to their criticism. This rate of refusal (around 20–25 workers refused) is in itself evidence of this limitation and indicates that the most vulnerable or the most apprehensive workers might be systematically not represented in the data.

Fourth, labour in tea gardens is context-specific due to residential enclosure, restricted mobility and intergenerational occupational assignment, which reduce transferability to other informal or low-wage environments. Therefore, the results are to be used with caution outside the plantation setting.

Fifth, the qualitative cross-sectional design will only record the perceived cycles and patterns without any temporal causal direction. Theoretical coherence eases the interpretive plausibility but cannot be used in place of longitudinal or experimental data.

Sixth, the effects of translation cannot be eradicated. Although the cross-checking procedures outlined in the methodology, cultural texture and tonal shade of the idioms of distress used by the Bangla still might be distorted partially even when translated to English even under the strict circumstances.

## Future research directions

The temporal validity of the pathways in this study would be enhanced by longitudinal qualitative studies that trace the changes in distress perceptions with payment events and seasonal changes (Prause et al., 2009). Mixed-method designs that would involve validated ERI and JD-R survey tools with qualitative stories would facilitate the quantification of exposure gradients and triangulation of mechanistic clarifications (Demerouti et al., 2001; Rugulies et al., 2017). A gender-specific study concerning the household bargaining patterns, care burden allocation and work dignity in terms of coping and mental health outcomes would broaden the incomplete results on Objective 2. It would be possible to assess the extent to which particular governance characteristics are most strongly related to worker distress through comparative estate-level studies addressing the variation in wage governance arrangements, including payment frequency, deduction structures and grievance access as well as management practices. The cross-national comparative studies that include tea estate workers in India, Sri Lanka and Bangladesh would aid in differentiating between country-specific regulatory implications, and the structural characteristics of the plantation labour shared by the region.

## Conclusion

This article has discussed how the tea garden workers in Sylhet and Moulvibazar view levels of wages and practices of payment governance as the determinants of their daily stress and psychological wellbeing. In individual, triadic and group interviews, participants described a uniform pattern of wage-related distress not only as an economic constraint but also as a complex psychosocial load, that is, as a continuous anxiety, worry, irritability and somatic discomfort.

This experience was found to have six dimensions. Wage insufficiency was seen as the main source of daily survival compromises, especially in food adequacy and children needs; workers noted that low wages paid on a weekly basis made structurally inaccessible regular meals and caused a certain degree of emotional instability. The time of payment and disruption was viewed as a stimulus of acute stress cycles in households because informal credit and crisis coping mechanisms were sought when the payment was delayed or decreased. This was increased by quota deductions and penalty practices that were seen as punitive and disproportionate: small productivity losses were converted into wage losses that were experienced as personally disheartening. The intersection between wage-related stress and care needs and social pressures related to children education and family respectability were narrated by women workers, and the fact is that the nominal wage equality cannot be transformed into the perceived burden. Dignity, voice and fair treatment were also manifested in terms of workers reporting unexplained deductions, denied complaints and lack of effective grievance systems, where powerlessness and emotional stress became mutually supportive of each other. Finally, employees expressed a set of similar priorities: fair pay, consistent food costs, punctual and clear pay, better ration quality and easy access to healthcare.

All these results add qualitative levels of knowledge to the comprehension of the tea garden wage system as a structural cause of mental health. They point out that reforms which should be made to alleviate distress and to allow more dignified working conditions should not only interact with the wage level but also with the quality of payment governance the predictability, transparency and equity with which wages are computed, reported and paid.

**Open peer review.** To view the open peer review materials for this article, please visit http://doi.org/10.1017/gmh.2026.10227.

**Data availability statement.** The information on the basis of this research is privately held due to the sensitivity of the information disclosed by participants and the non-disclosure agreements. Anonymised extracts are presented in the

manuscript. Prudent appeals to seek out additional details can be addressed to the respective author.

**Author contribution.** M.R.: Conceptualisation, methodology, data collection, formal analysis, writing – original draft, writing – review and editing. T.J.: Methodology, writing – review and editing, supervision.

**Financial support.** No funding was received.

**Competing interests.** The authors declare no conflicts of interest.

**Ethics statement.** The procedures used in ethics were based on the principles of voluntary participation, informed consent in written form, protection of privacy of the participant and minimisation of harm related to sensitive disclosures. Data collection was done with informed consent, which was in writing; there was no withdrawal penalty at any time. No clinical intervention was done.

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
