## [Reviewer Report]

Dear Authors, please find my detailed feedback aimed at refining and strengthening your manuscript for its next revision.

Title

The title does not fully capture what the study actually emphasizes in the findings, namely the role of payment governance such as deductions, quota penalties, delayed payments, and lack of transparency. The authors should consider whether the title should more explicitly reflect payment practices or wage governance rather than only low wages per se.

Abstract

The abstract presents contains several inconsistencies. The abstract refers to “12 participants” and then mentions additional group interviews for contextual triangulation, but it is unclear whether these group interviews are included in the analytic sample or treated as supplementary data. Later sections of the manuscript clearly exclude the Moulvibazar group interview from the analytic sample, which creates a mismatch between the abstract and methods. The abstract also implies a degree of explanatory or causal certainty by referring to “stress pathways,” which is problematic given the interpretivist, cross-sectional qualitative design. The language should be moderated to reflect that the findings represent participants’ experiences and interpretations rather than established causal mechanisms. Finally, the abstract would benefit from briefly stating how many major themes were identified and how they directly address the stated objectives of the study.

Impact Statement

The claims in the impact statement would be stronger if the authors clearly acknowledged that the study identifies perceived pathways and worker-articulated needs rather than evaluated interventions. Additionally, the statement could be improved by briefly distinguishing between short-term, actionable reforms and longer-term structural changes, mirroring the structure later used in the recommendations section.

Introduction

The introduction lacks structural clarity and scholarly precision. It interweaves subjective worker experiences, broad descriptive claims, and literature-based assertions without clearly distinguishing between them. This makes it difficult to identify what is empirically established versus what emerges from participant narratives. Several vivid claims, such as workers surviving on tea and bread for months or wages going unpaid for extended periods, are presented without clear attribution to either existing studies or interview data. These statements should either be cited or explicitly framed as participant-reported experiences.

Conceptually, key terms such as “payment governance” are not clearly defined early on. The introduction would benefit from a clearer progression that moves from context, to existing evidence, to an explicit research gap, and finally to the study’s objectives and contributions.

Objectives

The manuscript does not consistently demonstrate how each objective is addressed in the findings and discussion. In particular, the 2nd objective is only partially realized. While gendered caregiving burdens are discussed, systematic differentiation by employment status or job role is less evident. The 3rd objective is not convincingly supported by the methods or data presented. If such co-production did not occur in practice, the objective should be revised to reflect a synthesis derived from participant narratives.

Literature Review

The literature review is comprehensive but overly verbose. The authors should sharpen the review by clearly distinguishing between what is known about income volatility and mental health in general, what is known about plantation labor specifically, and what remains understudied at their intersection. There are also internal coherence issues. For example, the manuscript claims that few studies in Bangladesh link wage conditions directly to mental health, yet earlier sections cite studies documenting depression among tea workers. This dissimilarity needs to be clarified by specifying that prevalence studies exist, but mechanistic analyses of wage governance and mental distress are lacking.

Conceptual Model and Theoretical Framework

The manuscript overstates this framework by referring to hypotheses, which is not appropriate for an interpretivist qualitative study. The model should instead be framed as a sensitizing or guiding framework that informed data collection and analysis. Furthermore, while the discussion later connects themes to these theories, the manuscript would benefit from a clearer, earlier mapping between the conceptual model and the analytic themes. Explicitly showing how each theme corresponds to specific theoretical constructs would strengthen analytical transparency and theoretical contribution.

Methodology

The interpretivist qualitative design is clearly stated, but important details are missing or inconsistently reported. The timing of data collection, the languages used in interviews, and the translation process are not sufficiently described, despite later acknowledging translation as a limitation. The sampling description contains significant ambiguities, particularly regarding how individual, small-group, and group interviews are categorized and counted. The exclusion of the Moulvibazar group interview from the analytic sample is mentioned but not adequately justified or documented, including the absence of participant numbers for that group.

While the manuscript notes a high refusal rate, it does not sufficiently reflect on how this may have shaped the sample or findings. The data collection procedures are described partly in aspirational terms (“planned duration”), which should be replaced with actual practices where possible.

The authors describe their data analysis approach as reflexive thematic analysis while also emphasizing codebook development and double coding, which reflects a more positivist orientation. This apparent inconsistency should be resolved by clearly saying the analytic stance and purpose of multiple coders. Claims about saturation or information power should also be supported with clearer justification.

Findings

The findings rely heavily on a small number of participants, raising concerns about representativeness and balance. The manuscript would benefit from a clearer indication of how widely each theme was observed across participants and whether certain themes were more prominent among specific subgroups.

Some mechanisms, such as wage deductions tied to quota shortfalls, are described in ways that are difficult to interpret without further explanation of the wage calculation system. Distinct issues such as monetary deductions and in-kind ration quality are sometimes conflated and could be analytically separated for greater clarity. While gendered caregiving burdens are convincingly illustrated, the manuscript does not adequately demonstrate how these intersect with employment status or role differences, as outlined in the objectives. Observational material included in this section should be more clearly framed as contextual field notes rather than empirical evidence of mental health outcomes.

Discussion, Implications, and Recommendations

The discussion sometimes reiterates theory at the expense of deeper analytical integration. The strongest discussion would more directly link specific empirical themes to theoretical constructs, such as procedural injustice, resource loss, or effort–reward imbalance. The manuscript also needs to more clearly address variation across participant characteristics, rather than treating the workforce as homogeneous.

The policy and management implications would be stronger if the manuscript clarified which actors have the authority to implement different reforms and how these recommendations align with the study’s empirical findings.

Several recommendations are presented without sufficiently explicit grounding in the findings. Each proposed reform should be directly linked to specific themes or participant accounts to reinforce its empirical basis. Additionally, the language of co-production should be reconsidered unless the manuscript can clearly demonstrate participatory engagement beyond data collection.

Limitations and Future Research

The limitations section could be strengthened by more clearly acknowledging the small sample size, potential over-reliance on certain voices, and the likelihood that fear of retaliation constrained participant disclosures. The suggested future research directions could be expanded to include comparative studies across estates with different governance arrangements.

Conclusion

The conclusion occasionally uses language that overstates the findings, such as characterizing wage practices as “economic extortion.” Unless legally substantiated, such terminology should be replaced with more cautious academic language. The conclusion should consistently reflect the interpretive scope of the qualitative design and avoid causal assertions that exceed the evidence presented.

---

## [Reviewer Report]

This study serves as a poignant and necessary addition to the literature on the social determinants of mental health in Bangladesh. The authors have successfully utilized an interpretivist approach to map the “stress pathways” connecting wage governance to psychological distress among tea garden workers. The manuscript is well-written and socially relevant.

I recommend acceptance with minor revisions. These revisions are entirely focused on clarifying the existing methodology and refining the terminology used in the manuscript. No additional data collection is required.

Section-by-Section Review with Critiques

1. Title and Abstract

• Strength: The title is evocative and precise.

• Critique (Keywords): The keyword “Mental disability” appears inappropriate for this study. Unless the participants had clinically diagnosed cognitive or developmental disabilities, this term is misleading. The study describes stress, anxiety, and depression—which are “psychological distress” or “occupational stress.”

o Required Revision: Please replace “Mental disability” with a more accurate term like “Psychological Distress” or “Occupational Stress” in the keyword list.

2. Methodology (Clarification Only)

• Sample Size: The sample size is appropriate for a phenomenological study, but the justification in the text needs to be explicit to satisfy quantitative reviewers.

o Required Revision: Please add 1-2 sentences explicitly stating that data saturation was reached (i.e., that you stopped recruiting because no new themes were emerging from the interviews). This clarifies that the sample size was determined by the depth of data, not an arbitrary number.

• Interview Process: The abstract mentions “individual/small interviews.”

o Required Revision: Briefly clarify in the text what a “small interview” refers to (e.g., was it a dyadic interview with two people?). This is just a definition issue.

• Translation Transparency: The analysis relies on quotes translated from the local dialect.

o Required Revision: Please add a brief statement (2-3 sentences) explaining how you ensured the translations were accurate (e.g., “The authors, who are fluent in the dialect, cross-checked the English transcripts to ensure the emotional tone was preserved”).

3. Results / Analysis

• Theme Refinement: Some themes, such as “everyday survival” and “future anxiety,” appear to overlap.

o Required Revision: You do not need to change the data, but please sharpen the definitions of these themes in the text so the reader clearly sees the distinction between immediate survival (food/health) and long-term worry (children’s future).

4. Discussion

• Language Precision: At times, the text uses causal language (e.g., “low wages cause mental health issues”).

o Required Revision: Please soften this language to align with qualitative research standards (e.g., change to “low wages were perceived by participants as a primary driver of...”).

• Literature Context:

o Required Revision: Strengthen the discussion by adding a brief paragraph comparing your findings to existing literature on tea garden workers (e.g., in India or Sri Lanka). This highlights the relevance of your data without needing new findings.

---

## [Editor Report]

Dear Mr. Rabbani, 

Manuscript ID GMH-2026-0023 entitled “Counting Pennies, Carrying Stress: The Mental Health Impact of Low Wages on Tea Garden Workers in Sylhet and Moulvibazar,” which you submitted to Cambridge Prisms: Global Mental Health, has been reviewed. The comments of the reviewer(s) and editor(s) are included at the bottom of this letter. 

The manuscript is not acceptable for publication in its current form. However, I invite you to revise the manuscript in accordance with the reviewers' and the editor’s comments below and submit a revised version. 

When submitting your revised manuscript, you will respond to the comments made by the reviewer(s). In order to expedite the processing of the revised manuscript, please be as specific as possible in your response to the reviewer(s).

Once again, thank you for submitting your manuscript to Cambridge Prisms: Global Mental Health, and I look forward to receiving your revision. 

Sincerely, 

Dr Limkile Mpofu

Handling Editor, Cambridge Prisms: Global Mental Health

Comments to the Author

Dear Authors, please find my detailed feedback aimed at refining and strengthening your manuscript for its next revision.

Reviewer 1

Title

The title does not fully capture what the study actually emphasizes in the findings, namely the role of payment governance such as deductions, quota penalties, delayed payments, and lack of transparency. The authors should consider whether the title should more explicitly reflect payment practices or wage governance rather than only low wages per se.

Abstract

The abstract presents contains several inconsistencies. The abstract refers to “12 participants” and then mentions additional group interviews for contextual triangulation, but it is unclear whether these group interviews are included in the analytic sample or treated as supplementary data. Later sections of the manuscript clearly exclude the Moulvibazar group interview from the analytic sample, which creates a mismatch between the abstract and methods. The abstract also implies a degree of explanatory or causal certainty by referring to “stress pathways,” which is problematic given the interpretivist, cross-sectional qualitative design. The language should be moderated to reflect that the findings represent participants’ experiences and interpretations rather than established causal mechanisms. Finally, the abstract would benefit from briefly stating how many major themes were identified and how they directly address the stated objectives of the study.

Impact Statement

The claims in the impact statement would be stronger if the authors clearly acknowledged that the study identifies perceived pathways and worker-articulated needs rather than evaluated interventions. Additionally, the statement could be improved by briefly distinguishing between short-term, actionable reforms and longer-term structural changes, mirroring the structure later used in the recommendations section.

Introduction

The introduction lacks structural clarity and scholarly precision. It interweaves subjective worker experiences, broad descriptive claims, and literature-based assertions without clearly distinguishing between them. This makes it difficult to identify what is empirically established versus what emerges from participant narratives. Several vivid claims, such as workers surviving on tea and bread for months or wages going unpaid for extended periods, are presented without clear attribution to either existing studies or interview data. These statements should either be cited or explicitly framed as participant-reported experiences.

Conceptually, key terms such as “payment governance” are not clearly defined early on. The introduction would benefit from a clearer progression that moves from context, to existing evidence, to an explicit research gap, and finally to the study’s objectives and contributions.

Objectives

The manuscript does not consistently demonstrate how each objective is addressed in the findings and discussion. In particular, the 2nd objective is only partially realized. While gendered caregiving burdens are discussed, systematic differentiation by employment status or job role is less evident. The 3rd objective is not convincingly supported by the methods or data presented. If such co-production did not occur in practice, the objective should be revised to reflect a synthesis derived from participant narratives.

Literature Review

The literature review is comprehensive but overly verbose. The authors should sharpen the review by clearly distinguishing between what is known about income volatility and mental health in general, what is known about plantation labor specifically, and what remains understudied at their intersection. There are also internal coherence issues. For example, the manuscript claims that few studies in Bangladesh link wage conditions directly to mental health, yet earlier sections cite studies documenting depression among tea workers. This dissimilarity needs to be clarified by specifying that prevalence studies exist, but mechanistic analyses of wage governance and mental distress are lacking.

Conceptual Model and Theoretical Framework

The manuscript overstates this framework by referring to hypotheses, which is not appropriate for an interpretivist qualitative study. The model should instead be framed as a sensitizing or guiding framework that informed data collection and analysis. Furthermore, while the discussion later connects themes to these theories, the manuscript would benefit from a clearer, earlier mapping between the conceptual model and the analytic themes. Explicitly showing how each theme corresponds to specific theoretical constructs would strengthen analytical transparency and theoretical contribution.

Methodology

The interpretivist qualitative design is clearly stated, but important details are missing or inconsistently reported. The timing of data collection, the languages used in interviews, and the translation process are not sufficiently described, despite later acknowledging translation as a limitation. The sampling description contains significant ambiguities, particularly regarding how individual, small-group, and group interviews are categorized and counted. The exclusion of the Moulvibazar group interview from the analytic sample is mentioned but not adequately justified or documented, including the absence of participant numbers for that group.

While the manuscript notes a high refusal rate, it does not sufficiently reflect on how this may have shaped the sample or findings. The data collection procedures are described partly in aspirational terms (“planned duration”), which should be replaced with actual practices where possible.

The authors describe their data analysis approach as reflexive thematic analysis while also emphasizing codebook development and double coding, which reflects a more positivist orientation. This apparent inconsistency should be resolved by clearly saying the analytic stance and purpose of multiple coders. Claims about saturation or information power should also be supported with clearer justification.

Findings

The findings rely heavily on a small number of participants, raising concerns about representativeness and balance. The manuscript would benefit from a clearer indication of how widely each theme was observed across participants and whether certain themes were more prominent among specific subgroups.

Some mechanisms, such as wage deductions tied to quota shortfalls, are described in ways that are difficult to interpret without further explanation of the wage calculation system. Distinct issues such as monetary deductions and in-kind ration quality are sometimes conflated and could be analytically separated for greater clarity. While gendered caregiving burdens are convincingly illustrated, the manuscript does not adequately demonstrate how these intersect with employment status or role differences, as outlined in the objectives. Observational material included in this section should be more clearly framed as contextual field notes rather than empirical evidence of mental health outcomes.

Discussion, Implications, and Recommendations

The discussion sometimes reiterates theory at the expense of deeper analytical integration. The strongest discussion would more directly link specific empirical themes to theoretical constructs, such as procedural injustice, resource loss, or effort–reward imbalance. The manuscript also needs to more clearly address variation across participant characteristics, rather than treating the workforce as homogeneous.

The policy and management implications would be stronger if the manuscript clarified which actors have the authority to implement different reforms and how these recommendations align with the study’s empirical findings.

Several recommendations are presented without sufficiently explicit grounding in the findings. Each proposed reform should be directly linked to specific themes or participant accounts to reinforce its empirical basis. Additionally, the language of co-production should be reconsidered unless the manuscript can clearly demonstrate participatory engagement beyond data collection.

Limitations and Future Research

The limitations section could be strengthened by more clearly acknowledging the small sample size, potential over-reliance on certain voices, and the likelihood that fear of retaliation constrained participant disclosures. The suggested future research directions could be expanded to include comparative studies across estates with different governance arrangements.

Conclusion

The conclusion occasionally uses language that overstates the findings, such as characterizing wage practices as “economic extortion.” Unless legally substantiated, such terminology should be replaced with more cautious academic language. The conclusion should consistently reflect the interpretive scope of the qualitative design and avoid causal assertions that exceed the evidence presented. 

Reviewer 2

This study serves as a poignant and necessary addition to the literature on the social determinants of mental health in Bangladesh. The authors have successfully utilized an interpretivist approach to map the “stress pathways” connecting wage governance to psychological distress among tea garden workers. The manuscript is well-written and socially relevant.

I recommend acceptance with minor revisions. These revisions are entirely focused on clarifying the existing methodology and refining the terminology used in the manuscript. No additional data collection is required.

Section-by-Section Review with Critiques

1. Title and Abstract

• Strength: The title is evocative and precise.

• Critique (Keywords): The keyword “Mental disability” appears inappropriate for this study. Unless the participants had clinically diagnosed cognitive or developmental disabilities, this term is misleading. The study describes stress, anxiety, and depression—which are “psychological distress” or “occupational stress.”

o Required Revision: Please replace “Mental disability” with a more accurate term like “Psychological Distress” or “Occupational Stress” in the keyword list.

2. Methodology (Clarification Only)

• Sample Size: The sample size is appropriate for a phenomenological study, but the justification in the text needs to be explicit to satisfy quantitative reviewers.

o Required Revision: Please add 1-2 sentences explicitly stating that data saturation was reached (i.e., that you stopped recruiting because no new themes were emerging from the interviews). This clarifies that the sample size was determined by the depth of data, not an arbitrary number.

• Interview Process: The abstract mentions “individual/small interviews.”

o Required Revision: Briefly clarify in the text what a “small interview” refers to (e.g., was it a dyadic interview with two people?). This is just a definition issue.

• Translation Transparency: The analysis relies on quotes translated from the local dialect.

o Required Revision: Please add a brief statement (2-3 sentences) explaining how you ensured the translations were accurate (e.g., “The authors, who are fluent in the dialect, cross-checked the English transcripts to ensure the emotional tone was preserved”).

3. Results / Analysis

• Theme Refinement: Some themes, such as “everyday survival” and “future anxiety,” appear to overlap.

o Required Revision: You do not need to change the data, but please sharpen the definitions of these themes in the text so the reader clearly sees the distinction between immediate survival (food/health) and long-term worry (children’s future).

4. Discussion

• Language Precision: At times, the text uses causal language (e.g., “low wages cause mental health issues”).

o Required Revision: Please soften this language to align with qualitative research standards (e.g., change to “low wages were perceived by participants as a primary driver of...”).

• Literature Context:

o Required Revision: Strengthen the discussion by adding a brief paragraph comparing your findings to existing literature on tea garden workers (e.g., in India or Sri Lanka). This highlights the relevance of your data without needing new findings.